# Coronavirus risk perception and compliance with social distancing measures in a sample of young adults: Evidence from Switzerland

**Axel Franzen** ◉*, **Fabienne Wöhner** ◉

Institute of Sociology, University of Bern, Bern, Switzerland

◉ These authors contributed equally to this work.
* franzen@soz.unibe.ch

**Data Availability Statement:** The data is stored in the public repository of the University of Bern and available under https://boris.unibe.ch/151813/

## Abstract

The health risk of the coronavirus pandemic is age-specific. The symptoms of a COVID-19 infection are usually mild in the healthy population below the age of 65; however, the measures laid down to prevent the spread of the virus apply typically to the whole population. Hence, those who have a low risk of severe symptoms face a social dilemma in cooperating and complying with the safety measures: Cooperating in preventing the spread of the disease is good for society but comes with individual costs. These costs provide an incentive not to cooperate with the safety measures. In this paper we analyze via structural equation modelling a sample of young adults (N = 510) who were surveyed right after the end of the first lockdown period in Switzerland. We investigate why and to what extent they cooperated in preventing the epidemic by following the recommendation to stay at home as much as possible. We hypothesize that those respondents who perceive themselves to be personally at risk, or who have relatives belonging to the risk group, complied more often with the safety measures as compared to those without severe risks. Cooperating should also be linked to individuals' pro-social orientation. Furthermore, we hypothesize that those who believe that the virus is dangerous for society or who have a personal interest in protection show higher support for the general safety measures. Our empirical results show that compliance with the coronavirus social distancing measures was generally very high during the first lockdown. Although young adults perceived themselves to be at low personal risk, they still believed that the virus is dangerous for society. Those who had a personal interest in staying at home because they had relatives belonging to the risk group complied more often with the safety measures. Overall, the results suggest that the support of the preventive measures is the most important promoter of cooperation to prevent the spread of COVID-19.

## Introduction

The coronavirus pandemic arrived in Switzerland in February 2020, relatively soon after it had spread through neighboring Italy. The first Swiss COVID-19 death occurred in Lausanne on March 6, 2020. On March 16, the government of Switzerland declared a state of emergency:

**Funding:** The authors received no specific funding for this research

**Competing interests:** The authors have declared that no competing interests exist.

Among other measures, public meetings with more than 5 people were strictly forbidden. All shops, restaurants, schools, universities, and other public facilities had to close; exceptions were pharmacies, grocery stores, online-shops, and takeout restaurants. These measures lasted until April 27, and were subsequently relaxed. Hence, the first Swiss lockdown was intermediate in its severity, not as strict as the one in France, Italy or Spain but stricter that the one in Sweden. In particular, most companies in the service or industry sectors were allowed to stay open for business, and inhabitants could move freely outside their houses as long as physical distances of 2 meters or more were maintained.

By the end of 2020, the official coronavirus death toll in Switzerland had increased to about 7400 [1], and comparing the total number of deaths for 2020 (76,231) with the number the year before (67,515) or with other years reveals that there has been a substantive excess in deaths in 2020 [2]. However, the risk of dying from COVID-19 is age-specific. In Switzerland, as in most other countries, death from coronavirus occurs predominantly in the group of people who are 80 years or older (72%), and there are very few cases in the age groups below 65 (5%) [3]. Moreover, those who died from COVID-19 below the age of 65 mostly suffered from other health problems e.g. high blood pressure, cardio-vascular diseases, diabetes, chronic respiratory insufficiency, cancer or a weakened immune system [1].

In Switzerland 18.7% of the population belongs to the age group of 65 or above, and an estimated 14% below 65 suffers from relevant health problems [4, 5]. Hence, given that two thirds of the population does not belong to the group which is at severe risk of dying should they contract COVID-19, the interesting question is what did those who were not directly affected think about the crisis? How did they perceive the risk? To what extent did they approve of the political measures taken to prevent the spread of the infection, and how much did they comply with the social distancing measures?

For members of the risk group the decision to comply with the safety rules and to stay at home as much as possible potentially saves their lives. Hence, for at-risk individuals, complying with the safety regulations is self-beneficial and there is no incentive not to do so. However, the situation is different for the group not at risk. For individuals not directly experiencing a health risk the decision to comply with the safety regulations is like the decision to cooperate in a public good game [6, 7]. If they stay home, they are doing it predominantly for the benefit of others. If many stay home, the public good of interrupting the spread of the virus will be provided. However, each individual in this age group faces the temptation to freeride and to leave the provision of safety to others. Of course, if nobody complies with the restrictions, the infection rate will increase, possibly causing more deaths. But from the point of view of a single actor who is not directly affected by the risk, restricting the freedom of movement is a contribution to a public good.

In this study, we report the results of an online survey among a random sample of 510 young adults from the University of Bern, which was conducted in May 2020, immediately after the first lockdown was suspended in Switzerland. Hence, we study the behavior of individuals who are not directly affected in terms of health risks but who were asked to restrict their social activities for the common good. Our main research question is who complied with the measures to prevent the spread of COVID-19, and what are the most relevant drivers of cooperative behavior from those people mainly not at risk during the coronavirus pandemic.

The rest of the article is organized into 4 sections. In section "Theory and hypotheses" we review some studies that are similar to ours. Particularly, we review studies on vaccination, since the decision to vaccinate can also be conceptualized as the decision to contribute to a public good. Moreover, we take a look at recent studies that investigate the effect of differently framed appeals on participants' intention to engage in preventive measures to avoid the spread of COVID-19. At the end the section, we derive some hypotheses concerning the compliance

with the COVID-19 measures. Section "Data and methods" describes the survey and reports some descriptive results. The following section presents the result of a structural equation model which tests our hypotheses. Finally, in the last section, we discuss and summarize the main findings and suggest further research questions that emerge with the 'Corona Crisis'.

## Theory and hypotheses

In reaction to the coronavirus pandemic, on March 16, 2020 the Swiss government released a number of measures in order to prevent the spread of the virus. Some of them were recommended and voluntary, such as frequent disinfection of hands, and wearing masks; others were mandatory, such as the closure of public facilities (e.g. kindergartens, schools, and universities) and stores (with the exception of grocery stores or takeout restaurants). In comparison with other countries, the lockdown was less strict in Switzerland. Thus, it was recommended to stay at home whenever possible, but this was not mandatory. However, people were required to keep a distance of at least two meters and were not allowed to gather in groups of more than five people. Given that in Switzerland about two thirds of the population does not belong to the risk group, the interesting research question is why those people not at risk chose to comply with the measures. For those who are at risk, compliance with the safety measures is in their immediate self-interest. But for individuals who do not belong to the risk group, compliance with the coronavirus safety measures is like the decision to participate in vaccination programs against other infectious diseases [8]. Through vaccination, healthy individuals, who would only show mild symptoms if they were to catch the disease, protect mainly other individuals. Therefore, every individual recipient of a vaccination contributes to the extinction or containment of an infectious disease, as is the case with smallpox or MMR. Hence, the decision to participate in a vaccination program is like the decision to contribute to a public good. Generally, the literature on the provision of public goods has identified a number of conditions that increase individuals' willingness to contribute [9–14]. In particular, individuals contribute more often when the benefits of the public good are high and when the individual costs of contributing are low. Applied to the decision to participate in vaccination this means that individuals are more willing to participate if the consequences of the disease are severe, or put differently, if the benefits of extinction of the disease are great, and if the individual health risks associated with the vaccine are low [15–19].

Recent studies have applied the public good framework to the intention of complying with safety measures in the COVID-19 pandemic [20–25]. For example, Jordan et al. investigate in a series of studies using Amazon Mechanical Turk the effect of differently framed text messages on participants' intention to engage in various safety measures (e.g. washing hands, or avoiding to socialize with others). The messages either highlighted that the virus is a personal threat or a threat for the community. They found that the prosocial frame increased participants' intention to engage in preventive behaviors more or at least as well as the personal frame as compared to a control group [20]. Similar results are also reported in related studies that also use Amazon Mechanical Turk and investigate the intention to wear a face mask [21, 22]. However, some studies also present mixed results. For instance, Banker and Park presented three different ads on Facebook "protect yourself" "protect your loved ones", and "protect your community". They found that the ads referring to self and loved ones elicited slightly higher clickthrough rates than the community frame [23]. A study by Heffner et al. investigates pro-social framed messages versus threat-related messages on participants' willingness to self-isolate. They find that both types of messages perform equally well [26].

While most former studies focus on individuals' intention to comply with the Corona safety measures, we study individuals' actual behavior. Drawing on the results of former studies we

formulate a number of hypotheses of why individuals are expected to conform to the safety measures to avoid the spread of COVID-19. First, following key results from vaccinations studies, we hypothesize that compliance with the coronavirus measures increases with increased acceptance of the safety measures (H1). If the safety measures are evaluated as meaningful, efficient, and safe then compliance with the coronavirus measures should increase.

Second, individuals' subjective risk perception that the virus is personally harmful for them should increase compliance with coronavirus safety measures (H2). For individuals with a health risk complying with the safety measures is directly self-beneficial. Additionally, the perception that the virus is personally harmful should also increase the general acceptance of safety measures (H3) because these measures supposedly limit the spread of the virus and thereby the individual risk of becoming infected. Thus, individual risk perceptions should have a direct effect on compliance with the safety measures as well as an indirect effect via the acceptance of the measures.

Moreover, even when individuals are not personally at risk they may still have relatives and friends that belong to the risk group. The closer the contact to members of the risk group, the stronger should be the motive for complying with the measures (H4). Additionally, concern for friends and family should also increase the acceptance of coronavirus safety measures, since these protect friends and relatives (H5). For individuals who are not personally at risk, and also have no close contacts to members of the risk group, compliance with the coronavirus measures resembles the decision to contribute to a public good. The necessity to do so should become more understandable and convincing if the virus is perceived as being harmful to society. Hence, the more the virus is perceived as threatening to others in society, as compared to being less or not at all dangerous, the stronger the motive for accepting the public safety measures (H6). In contrast to the individual risk perception, the perception that the coronavirus is dangerous for society (social risk) should not necessarily increase compliance with the safety measures since compliance with the measures is a public good, and also since—in the absence of a personal risk—incentives to contribute to the public good are lacking.

Much laboratory research in game theory and on public good provision has demonstrated that not all individuals behave in the same way. Some individuals show more empathy [27, 28] than others, or show a stronger prosocial orientation [29, 30]. Similarly, we expect that individuals with a higher prosocial attitude comply more often with the coronavirus measures than individuals with a stronger pro-self orientation (H7). Also, research on public good provision suggests that individuals comply more often when they are embedded in social networks that reward prosocial behavior or sanction uncooperative behavior [14, 31]. Hence, the opportunity to gain social approval (or to avoid disapproval) might be a reason for individuals to comply with COVID-19 measures. However, in the case of coronavirus measures compliance is not easily visible or obvious. One of the measures recommends staying at home as much as possible; hence, leaving the home is legitimate if necessary, and it is exactly the nature of this necessity that is not visible. Thus, uncooperative behavior is not detectable by social contacts making the motive of social approval-seeking less applicable in this case.

Moreover, much research on health-related behavior has found that women are generally more risk averse and more health conscious then men [32, 33]. This also applies to compliance with measures against pandemics [34] and to social distancing in the coronavirus pandemic [35, 36]. Hence, we generally expect that women more often comply with the measures as compared to men (H8). Taken together, our theoretical considerations can be summed up by the model depicted in Fig 1. Since the theoretical model contains manifest variables (indicated by a rectangle), which are measured directly by a single indicator, as well as latent constructs (indicated by an oval), which are measured by multiple indicators, we employ structural

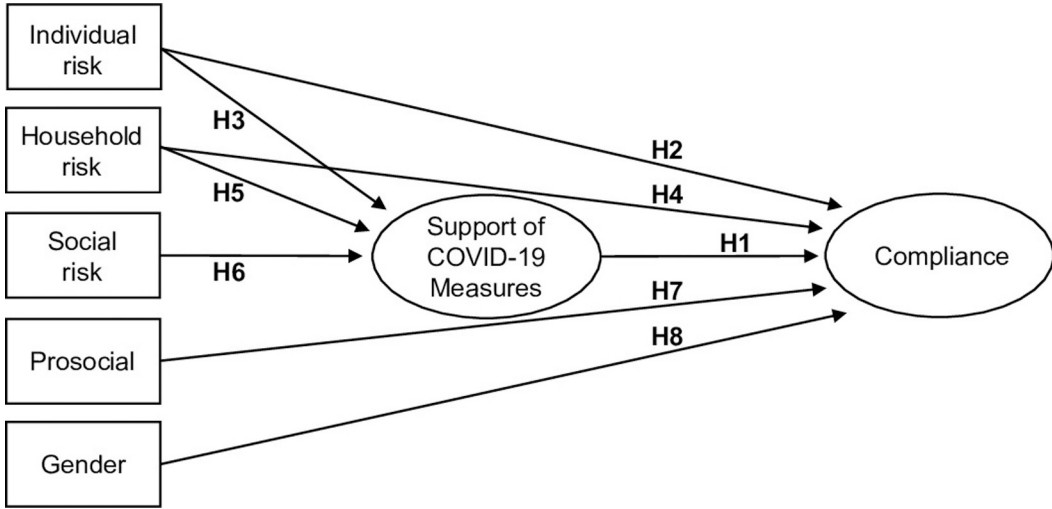

**Fig 1. The hypothesized determinants of complying with coronavirus social distancing measures.**

equation modelling to test the hypotheses. Before the result of such a model is presented, we next describe the data and the measurements of the variables.

## Data and measurements

The original purpose of our study was to test the reliability and validity of various measurement instruments. For this reason, we conducted a two-wave panel survey of a random sample of 510 students enrolled in a regular BA or MA program at the University of Bern. Ethical clearance was obtained by the Ethics Committee of the Faculty of Business Administration, Economics and Social Sciences of the University of Bern. The first wave took place between the end of February 2020 and middle of March 2020 and the second wave between end of April 2020 and end of May 2020. The coronavirus safety measures were implemented in Switzerland on March 16, 2020, just after we finished the first wave, and lasted until April 27. Hence, we were able to incorporate questions on the perception of the coronavirus epidemic into the second wave, which we started right after lockdown ended. In this paper we only report the findings that result from the second wave of the survey. For purposes that are not relevant for this study about half of the interviews were conducted face-to-face either personally or via online communication tools (e.g. Skype or Zoom). Besides questions on the coronavirus, the questionnaires of the first and second wave were mostly identical. Both waves contained around 50 questions, and lasted on average for about 25 minutes. Since we did not find any mode effects, in what follows we report the results of both interviewing modes. Because most questions on what the survey participants think about the coronavirus safety measures or what they did during the lockdown are retrospective, it is very important that the time lag between the lockdown period and the interviews is as short as possible. Hence, we were fortunate to be able to integrate the questions about the coronavirus crisis into the second wave of an ongoing study right after the end of the lookdown. This makes this data particularly valuable for a test of our hypotheses.

For our analyses, we excluded respondents older than 40 years (N = 5), since these cases are untypical for the student population. Also, people with a gender other than male or female were excluded from our analyses due to the very small number of three observations. In order to be able to detect respondents who did not answer the questionnaire thoroughly, we included

a fake question in the online version of the survey. This question simply contained the instruction that none of the given answer categories should be ticked. Hence, respondents who nonetheless ticked an answer category in this question either did not read this and possibly other questions very carefully or ignored the instructions. Also, these cases were excluded from further analyses (N = 9). Overall, these exclusions result in a valid sample of 493 observations. However, these exclusions did not affect our results. Mean age of the survey participants is 23.6, and 65% are female. Hence, females are slightly overrepresented in our survey as compared to the student population of the University of Bern (57% female). But other socio-demographic variables (e.g. subject areas, proportion enrolled in Bachelor and Master) resemble the distribution of the population.

We conceptualize compliance to the coronavirus safety regulations as a latent variable, and measure it according to three indicators. First, on a 5-point Likert scale ranging from "not at all" (1) to "very strictly" (5), participants were asked how much they complied with the recommendation to stay at home as much as possible. The second indicator is the question of whether respondents made occasional exceptions to staying at home. The question has five answer categories ranging from "never" (1) to "very often" (5), and hence, is coded in the opposite direction as compared to the first indicator, balancing the index. For the statistical analysis, the coding of the answer categories is reversed. Thirdly, we asked how many friends or relatives participants met in their leisure time during the week before the interview. Meeting a large number of friends is not conforming to the coronavirus measures and hence indicates non-compliance to the safety measures. For the analyses, we took the natural logarithm of the number of people, to achieve a more similar range and variance compared to the other two 5-point scales used for the construction of the latent compliance variable. Overall, 83% of the respondents report that they complied most of the time or very strictly with the recommendation to stay at home. Accordingly, 55.6% said that they never or rarely made exceptions, and the average number of people met in the week before the interview was 4.9 as compared to 9 in the first wave of the survey before the lockdown, which included the same question. Hence, the answers to all three indicators demonstrate that participants did show high levels of compliance with the social distancing measures.

Next, we quantify the support of the coronavirus measures using the acceptance of 12 different single measures put into practice by the Swiss government. Fig 2 lists the 12 different measures. For each measure respondents were asked to what extent they agree with the measure on a five-point answering scale ranging from "not at all" (1) to "very much" (5). Fig 2 depicts the percentage of respondents agreeing or agreeing very much with a measure. As can be seen most measures were highly accepted. This is particularly true for washing hands thoroughly (99%) and keeping social distance (93%). High acceptance can also be observed for the closing of restaurants and bars (84.7%), and universities (79%). Wearing a face mask received very little support in Switzerland (28%), which reflects the ambivalent public discussion in Switzerland about the preventive effect of masks during the first lockdown.

As hypothesized, compliance with the coronavirus measures should be driven by acceptance of the safety regulations, which should be directly affected by the individual and social risk perception. We measured individuals' risk perceptions by asking how dangerous respondents think a coronavirus infection would be for themselves personally on an 11-point answering scale ranging from "not at all dangerous" (0) to "extremely dangerous" (10). Similarly, the social risk perception was measured by asking how dangerous respondents believe the coronavirus epidemic is for the health of the Swiss population. The distribution of the two variables is depicted in Fig 3. As can be seen from Fig 3, most respondents perceive a low individual risk (median = 2) but a high social risk (median = 6). The descriptive results confirm previous findings [37] and reflect the fact that we have a sample of young adults for whom the virus is

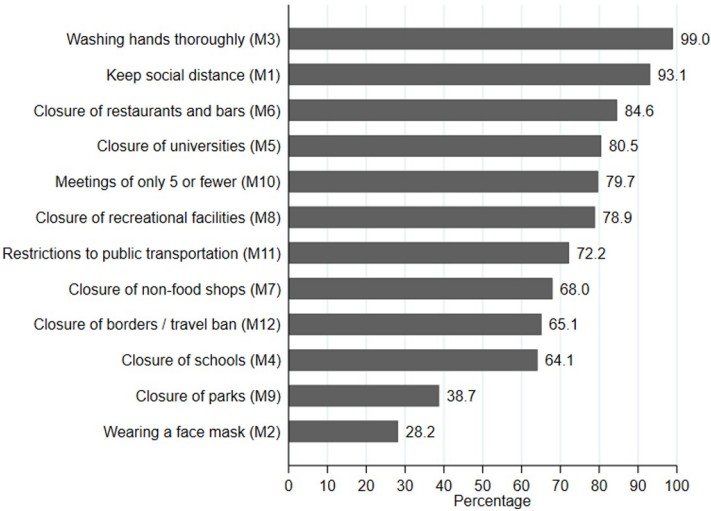

**Fig 2. Acceptance of coronavirus measures.** N = 493. Each measure was surveyed via five point Likert scales ranging from "do not agree at all" (1) to "agree very much" (5). The figure displays the proportion of respondents supporting a measure weakly or strongly (categories 4 and 5).

mostly harmless. A further reason to conform to the safety measures is if subjects live together in one household with individuals who belong to the risk group. This might apply in our sample to students who still share the household for instance with their parents or other family members who might be older or suffer from health risks. In our sample 27% of respondents report that they live together with at least one at-risk person.

Finally, our theoretical model proposes that individuals with a prosocial attitude should conform more strictly to the coronavirus measures as compared to individuals who are less prosocial. We measure pro-sociality via a revealed preference approach. Participants received 20 Swiss Francs (about $20) for participation in the survey. At the end of the questionnaire, we asked respondents if they wished to donate some of their payment to a charitable organization. A relatively large proportion of our respondents (58%) decided to donate some money to a

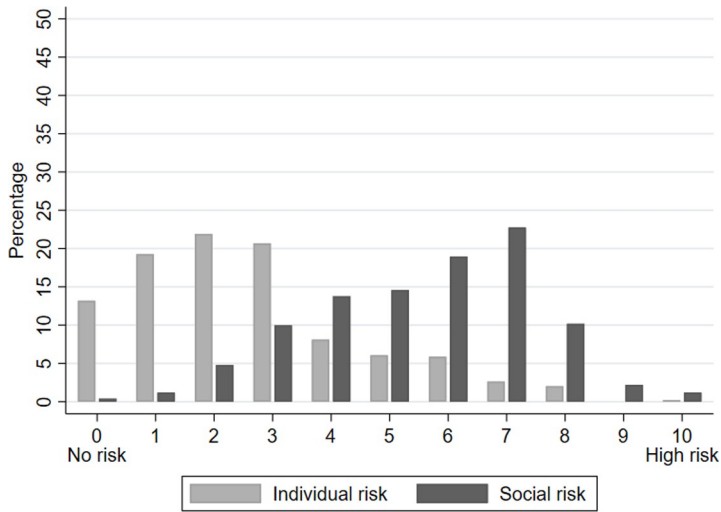

**Fig 3. Perceived individual and social risk.** N = 493.

charitable organization. Respondents who chose the option of donating were then asked how much and to which organization they wanted to donate. We listed some environmental organizations (Greenpeace, World Wide Fund for Nature WWF) but participants also had the chance to name an organization of their own choice. Those who decided to donate some money are classified as pro-social individuals. Descriptive information of all variables are depicted in S1 Table.

## Results

Since some of our variables can be conceptualized as latent variables, we test hypotheses H1 through H8 via a structural equation model [38–40]. We estimated the model using the software program Stata 16 [41–43]. Since our variables are neither normally distributed nor follow a multivariate normal distribution, we use maximum likelihood estimation applying the Satorra-Bentler correction for standard errors and model fit parameters [43–45]. We regressed all exogenous variables on both dependent latent constructs. Modification indices suggested the addition of an error covariance between $M_4$ (closure of schools) and $M_5$ (closure of universities), which are both indicators of the latent construct of the support of measures. The inclusion of this covariance results in an improvement of the model fit ($\chi^2 = 97.7$, df = 1, p < 0.01), and makes sense from a theoretical perspective, since both indicators are addressing educational facilities. The Satorra-Bentler corrected $\chi^2$ value of the model is 338 with 166 degrees of freedom. However, because of the large sample size and the relatively large number of indicators the $\chi^2$-statistic is not an appropriate test statistic for this model [46–48]. Following the literature, we focus on CFI and TLI instead as goodness-of-fit statistics. In our model the Satorra-Bentler corrected CFI statistic results in a value of 0.93 and the TLI goodness-of-fit statistic in a value of 0.91. Both exceed the minimal threshold of 0.9 indicating an acceptable model fit. The SRMR fit statistic has a value of 0.042, which is smaller than 0.05 suggesting also a good model fit. A further test statistic is the RMSEA. In our case, RMSEA has a value of 0.049 with a 90% confidence interval (CI) of 0.042 to 0.056. Hence, also the RMSEA indicates a good model fit. Additionally, the test of close fit—testing that RMSEA is smaller than 0.05—is statistically non-significant (p = 0.65), as desired. Overall, the different fit statistics indicate an adequate to good fit of our theoretical model to the observed data.

The unstandardized results of the estimation are depicted in Fig 4. The three indicators "staying at home", "making exceptions", and the "number of meetings with friends" are all making statistical significant contribution to the measurement of the latent construct "Compliance". Similar conclusions can be drawn for the measurement of the latent construct "Support of COVID-19 Measures", which we measure using the 12 indicators that are listed in S1 Table. All indicators contribute statistically significantly to the measurement of this latent variable. However, some of the 12 indicators only make small contributions to the measurement of the COVID-19 measures. Therefore, we also ran a model in which we excluded all indicators with low factor loadings (e.g. M2, M3, M11, and M12). The exclusion of these indicators improved the model fit ($\chi^2 = 203$, df = 96, CFI = 0.952, TLI = 0.939, SRMR = 0.035, RMSEA = 0.048). However, neither the results of our structural coefficients nor their standard errors changed substantially. Because there was no difference in the results, and due to the theoretical interest in all different COVID-19 measures, we decided to report the complete model despite its slightly lower fit.

Next, we discuss the empirical results with respect to the hypotheses. The latent variables are coded according to the coding of the indicator that is restricted to 1. Hence, in our case both latent constructs have a range from 1 (low compliance, low support) to 5 (strict compliance, high support). Consequently, a one-unit increase in the Support of Measures leads to a 0.45 unit increase in Compliance (see Fig 4). The standardized coefficient of this effect is 0.44.

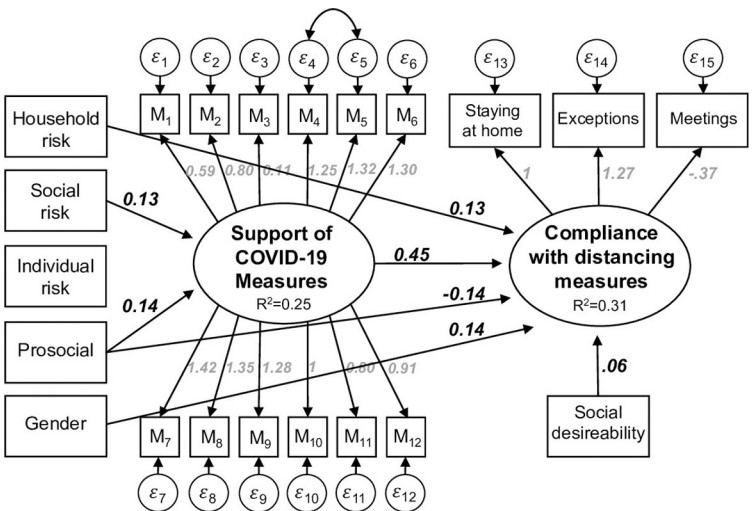

**Fig 4. A structural equation model of the compliance to corona measures.** N = 493. All reported coefficients are unstandardized and statistically significant at least at the 5% level.

Thus, the support of coronavirus measures has the strongest impact on Compliance among all considered variables. Therefore, individuals who are in favor of the measures also show strong compliance with them. This result confirms hypothesis H1.

The effects of individuals' risk perception (individual risk, household risk and social risk) are somewhat ambivalent. According to our results, respondents' perception of their own individual risk has neither a statistically significant effect on the acceptance of the measures nor on complying with them. Hence, hypotheses H2 and H3 are not supported by our data. However, individuals who live in one household with members of the risk group do show higher compliance with the measures, e.g. reported to stay home more often. However, the effect of living with a member of the risk group is relatively weak (0.13). In contrast, no statistically significant effect was found of living with a member of the risk group on the acceptance of the measures. Therefore, hypothesis H4 does receive empirical support but hypothesis H5 does not. As expected, individuals' perception of the social risk of the coronavirus has a positive impact on the acceptance of the measures (0.13), which supports hypothesis H6. This effect is also quite substantial since the social risk is measured on an 11-point scale. Put differently, a one standard deviation increase in the perceived social risk (which corresponds to approximately 2 points on the 11-point scale), leads to a 0.44 standard deviations increase on the support of measures. Prosocial orientation has an ambivalent effect: prosocial individuals (measured by donations) accept the measures more often (0.14), but comply to them less strictly (-0.14). This result actually contradicts hypothesis H7, which assumes that a prosocial orientation should also increase compliance. Finally, hypothesis H8, which states that women comply more often than men with the coronavirus measures, is supported by our data, but the effect is relatively small (0.14).

Since our data is based on a survey of self-reported behavior, socially-desirable answering needs to be considered. To control for such influences, we measured social desirability according to the well-known Marlowe-Crowne Scale [49, 50] and incorporate it into the model. As expected, the results show that individuals with higher levels of social desirability also report behavior more compliant with the measures. However, the effect (0.06) is relatively small (in terms of standard deviation 0.2). Overall, the model explains 31% of the variance of the social distancing compliance and 25% of the variance of the acceptance of the measures.

## Summary and discussion

In this study we formulated eight hypotheses specifying the reasons why young adults should accept the safety measures to prevent the spread of the coronavirus and which factors are driving social distancing compliance. The empirical model supports hypothesis H1: Individuals who are in favor of the coronavirus measures also conform to them, e.g. they stay home, rarely make exceptions and limit meetings with friends and relatives. This effect of the acceptance of the measures on compliance with them is the strongest and most important one in our model. Surprisingly, individuals' personal and social risk perceptions do not matter for the degree of compliance; hence, the empirical analysis refutes H2 and H3, which confirms also findings by Moussaoui et al. [36]. Another important finding is that the perceived risk of the coronavirus to society increases the acceptance of the different safety measures (H6). This effect is also very strong. Living together in one household with people who belong to the risk group increases compliance, lending support to H4. Prosocials support the safety measures, which we did not expect, but do not comply more strictly, as we expected, refuting hypothesis H7. Finally, we find that women comply somewhat more with the measures than do men.

In summary, our study shows that acceptance of the coronavirus safety measures was very high for young adults during the first lockdown period in Switzerland and this acceptance dominates the compliance with the coronavirus measures. This result is surprising. After all, for young and healthy individuals complying with the coronavirus social distancing measures is like contributing to a public good. Much research has shown that in other areas contributions to public goods are typically low. For instance, research on the determinants of environmental behavior consistently shows that individuals have high levels of environmental concern but do not act in accordance with their attitudes [51, 52]. Gaps between attitudes and behavior are also found in health-related behavior. Individuals often have the goal of living healthily but the associated overt action (good nutrition, sport) lags substantially behind [53–55]. Our results demonstrate that such a gap between attitudes and behavior was basically absent during the first lockdown in Switzerland. The emergency situation obviously induced many to comply with their convictions to stay at home as much as possible even if this behavior is costly and not self-beneficial.

Our research also has a number of limitations and raises many questions for further research. First of all, the first lockdown was relatively short in Switzerland and it was not particularly strict. Hence, the freedom to leave the house and move around was not restricted. In this situation maybe acceptance was also high because individuals in Switzerland were well aware that the lockdown was much stricter in the rest of Europe, particularly in neighboring Italy and Germany. Furthermore, the first lockdown was short in Switzerland and lasted only 6 weeks. Hence, it's possible that the high acceptance of the governmental safety measures might erode if lockdowns were to last longer [56]. Moreover, in Switzerland governmental institutions generally enjoy high levels of trust. Hence, it would be interesting to compare compliance with lockdown measures in Switzerland with compliance in other countries where lockdown lasted longer, was stricter, and where governments perhaps experience less trust. It would also have been interesting to investigate a random sample of the whole population instead of a sample of young adults. Random population samples would probably have generated larger variance of key variables such as the perceived individual and social risk. Also, we have investigated basically the conformity to injunctive norms, e.g. to the norm that the safety measures should be adhered to. However, many research areas in sociology also pay attention to descriptive norms, e.g. the perception of what other people do [24]. Unfortunately, questions of what the respondents think of what others do and how others react to the crisis were not included in this study. Finally, our analysis is based on a survey of self-reported behavior

which is knowingly biased by social desirability. We did control for this by measuring and incorporating social desirability; however, the control is not perfect and most likely does not eliminate the bias completely. Future research could vastly profit by using nonreactive compliance measures such as GPS-based mobility profiles of the sample under scrutiny. Since we measured attitudes and self-reported behavior at the same time, we also cannot exclude the possibility that subjects adopted their attitudes to their self-perceived behavior instead of the other way round. Hence, the assumed causal attitude-behavior link should be investigated by non-intrusive behavioral measures in combination with a time lag of the observed behavior.

## Supporting information

**S1 Table. List of variables.**
(DOCX)

## Author Contributions

**Conceptualization:** Axel Franzen.

**Data curation:** Fabienne Wöhner.

**Formal analysis:** Axel Franzen, Fabienne Wöhner.

**Investigation:** Axel Franzen.

**Methodology:** Axel Franzen, Fabienne Wöhner.

**Project administration:** Axel Franzen.

**Supervision:** Axel Franzen.

**Visualization:** Fabienne Wöhner.

**Writing – original draft:** Axel Franzen.

**Writing – review & editing:** Axel Franzen, Fabienne Wöhner.

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
