## [Decision Letter · Decision Letter 0]

26 Jan 2021

PONE-D-20-38429

Coronavirus risk perception and compliance with social distancing measures in a sample of young adults: Evidence from Switzerland

PLOS ONE

Dear Dr. Franzen,

Thank you for submitting your manuscript to PLOS ONE. After careful consideration, we feel that it has merit but does not fully meet PLOS ONE’s publication criteria as it currently stands. Therefore, we invite you to submit a revised version of the manuscript that addresses the points raised during the review process.

Please find below the reviewers' comments, as well as those of mine.

We look forward to receiving your revised manuscript.

Kind regards,

Valerio Capraro

Academic Editor

PLOS ONE

Journal Requirements:

Additional Editor Comments:

I have now collected two reviews from two experts in the field. Both reviewers like the paper and think that it can be published after a minor revision. I agree with them, therefore I would like to invite you to revise your work for Plos One. On top of the reviewers' comments, I would like to add a few more comments, mainly regarding the literature review, which, in my opinion, fails to acknowledge several related papers. (i) The "perspective article" on what social and behavioural science can do to support pandemic response, published by Van Bavel et al. in Nature Human Behaviour can be a useful general reference for the introduction; (ii) The correlation between perceived risk and intentions to engage in some preventative measures (social distancing and mask wearing) was already reported by Capraro & Barcelo (2020); (iii) The fact that pro-social messages, in particular messages highlighting that the coronavirus is a threat to the community, increases intentions to engage in some preventative measures (mask wearing) was also already reported by Capraro & Barcelo (2020) - note, however, that the same message does not increase intentions to practice social distancing (see also Jordan et al. 2020); (iv) Finally, it's missing a detailed review of the emerging literature on what messages increase pandemic response. I am aware of the following studies: Banker & Park, 2020; Bilancini et al. 2020; Capraro & Barcelo, 2020; Capraro & Barcelo, in press; Heffner et al. 2020; Jordan et al. 2020; Lunn et al. 2020; Pfattheicher et al. 2020. Of course, it is not a requirement to cite all these works, but I am mentioning them because they look very related to your manuscript.

References

Banker, S., & Park, J. (2020). Evaluating prosocial COVID-19 messaging frames: Evidence from a field study on Facebook. Judgment and Decision Making, 15(6), 1037-1043.

Bilancini E, Boncinelli L, Capraro V, Celadin T, Di Paolo R (2020) The effect of norm-based messages on reading and understanding COVID-19 pandemic response governmental rules. Journal of Behavioral Economics for Policy 4, Special Issue 1, 45-55.

Capraro, V., & Barcelo, H. (2020). The effect of messaging and gender on intentions to wear a face covering to slow down COVID-19 transmission. Journal of Behavioral Economics for Policy, 4, Special Issue 2, 45-55.

Capraro, V., & Barcelo, H. (In press). Priming reasoning increases intentions to wear a face covering to slow down COVID-19 transmission. Applied Cognitive Psychology.

Heffner, J., Vives, M. L., & FeldmanHall, O. (2020). Emotional responses to prosocial messages increase willingness to self-isolate during the COVID-19 pandemic. Personality and Individual Differences, 170, 110420.

Jordan, J., Yoeli, E., & Rand, D. (2020). Don’t get it or don’t spread it? Comparing self-interested versus prosocially framed COVID-19 prevention messaging. https://psyarxiv.com/yuq7x

Lunn, P. D., Timmons, S., Barjaková, M., Belton, C. A., Julienne, H., & Lavin, C. (2020). Motivating social distancing during the Covid-19 pandemic: An online experiment. Social Science & Medicine, 113478.

Pfattheicher, S., Nockur, L., Böhm, R., Sassenrath, C., & Petersen, M. B. (In press). The emotional path to action: Empathy promotes physical distancing during the COVID-19 pandemic. Psychological Science.

Van Bavel, J. J., et al. (2020). Using social and behavioural science to support COVID-19 pandemic response. Nature Human Behaviour, 4, 460-471.

Reviewers' comments:

Reviewer's Responses to Questions

**Comments to the Author**

1. Is the manuscript technically sound, and do the data support the conclusions?

Reviewer #1: Partly

Reviewer #2: Yes

2. Has the statistical analysis been performed appropriately and rigorously? 

Reviewer #1: Yes

Reviewer #2: Yes

3. Have the authors made all data underlying the findings in their manuscript fully available?

Reviewer #1: Yes

Reviewer #2: Yes

4. Is the manuscript presented in an intelligible fashion and written in standard English?

Reviewer #1: Yes

Reviewer #2: Yes

5. Review Comments to the Author

Reviewer #1: The paper reports data on compliance with social distancing measures in young adults. I acknowledge that new findings on this topic are being published at a rapid rate so it is difficult to keep up, but some relevant publications are missing. The authors introduce the pandemic as a social dilemma situation, which have been mentionned earlier by Ling Hoh Teck & Chyong, 2020 Effects of the Coronavirus (COVID-19) Pandemic on Social Behaviours: From a Social Dilemma Perspective. Technium Social Sciences Journal, 7, 312–320. Another related paper is by Moussaoui, Ofosu & Desrichard (2020) Social Psychological Correlates of Protective Behaviours in the COVID‐19 Outbreak: Evidence and Recommendations from a Nationally Representative Sample. Appl Psychol Health Well Being: 1183-1204. doi: 10.1111/aphw.12235. The authors also consider the social dilemma perspective in relation to the adoption of protective measures. Moreover, some results reported in the reviewed paper replicates results by Moussaoui et al. (notably the fact that personal risk (susceptibility) is not associated with compliance). There are probably other recent publications about determinants of compliance with protective measures and authors should make sure they include all relevant publications, and integrate their results into the broader literature.

I think that the authors should consider the causality dimension in their discussion of their results. It is implied that it is the support that induces compliance, but as both were measured at the same time, the other way around cannot be ruled out. This seems especially plausible as the lockdown was over at the time of data collection, and people could rationalize their behaviour by increasing the support for the measures and so justify their behaviour. The method cannot provide support for one or the other option, so this needs to be discussed as a limitation of the study.

There are some sentences in the introduction that are questionnable in my perspective, and do not seem essential to the paper. Notably l.114-116 on the "no substantial excess deaths" due to the pandemic. I think this simple temporal comparison misses a lot of contextual elements, like the probably reduced number of deaths related to acidents for example, due to lockdown. So I am not sure that the claim is accurate, and as it is not essential to the paper I would remove it. Similar comment for the sentence l.120-123 on the fact that those who died from COVID were suffering from other heatlh problems. I am again not sure about the validity of the sentence. I would suggest to remove the sentence, or to provide scientific support for it such as medical publications.

p.16: "desireability" should be corrected (3 times)

the figures do not seem to be of high-enough resolution

the N in the abstract (N=500) do not match the N in the paper (N=510) (but 500 is also mentionned l.143 so I'm unclear about that).

Reviewer #2: This is a well-written paper studying the self-reported compliance with, and support of, Swiss COVID-19 safety measures among university students. Overall, I found the results to be interesting (and some surprising). Since the analysis is based on a single survey, the authors are not able to identify a causal relationship between support for, and compliance with, COVID-19 measures. Thus, the results presented here would not be sufficient to guide policy choices or evaluate their efficacy. However, estimating associations between individual attributes, support for COVID-19 measures, and compliance is a useful descriptive exercise in that it provides us with a idea of how an important segment of society — those at low risk — feel about government efforts that require voluntary cooperation. Thus, I support publication. However, the causal language used to describe relationships should be altered to reflect that they are only correlational.

6. PLOS authors have the option to publish the peer review history of their article (what does this mean?). If published, this will include your full peer review and any attached files.

Reviewer #1: No

Reviewer #2: No

---

## [Author Response · Author response to Decision Letter 0]

5 Feb 2021

Dear Valerio Capraro, 

First of all, we thank you and the two reviewers very much for the effort of reviewing our manuscript and for the helpful suggestions. We incorporated all of them and respond to every comment point by point as they were made. 

Response to the editor

The editor recommends a list of references to be included in the paper. We thank the editor for these helpful references. We incorporated all of them into the paper, with only one exception. Particularly, the revised paper contains a new paragraph referring to this recent body of literature. This paragraph as well as the new references are marked in the marked-up copy of the manuscript. 

Response to reviewer 1

1.) The reviewer suggests to include two further references: Hoh Teck Ling & Mee Chyong Ho 2020, and Moussaoui et al. 2020. Both suggestions were very helpful and we did incorporate them into the paper. The new references are marked in the marked-up copy of the manuscript. Furthermore, we point out in the summary and discussion part, that our finding replicates the results from Moussaoui et al. 2020 (see marked up copy of the manuscript). Moreover, we updated the literature review as was also suggested by the editor. 

2.) We agree with the reviewer that we should be careful with causal implications. Therefore, we included a statement in the discussion part saying “Since we measured attitudes and self-reported behavior at the same time, we also cannot exclude the possibility that subjects adopted their attitudes to their self-perceived behavior instead of the other way round. Hence, the assumed causal attitude-behavior link should be investigated by non-intrusive behavioral measures in combination with a time lag of the observed behavior.” 

3.) The reviewer questions the paragraph on the number of Corona deaths in the introduction. In the revised version we updated all numbers according to the official statistics in Switzerland. As it turns out, Switzerland did experience excess deaths due to the Corona Pandemic by the end of 2020. Accordingly, we wrote in the revised manuscript: “By the end of 2020, the official coronavirus death toll in Switzerland had increased to about 7400 [1], and comparing the total number of deaths for 2020 (76,231) with the number the year before (67,515) or with other years reveals that there has been a substantive excess in deaths in 2020 [2].” We hope that this takes care of the reviewer’s concerns. 

4.) Concerning the statement about health problems we did provide a reference in the revised manuscript: FOPH. Weekly situation report on the epidemiological situation in Switzerland and the Principality of Liechtenstein: Week 53. Federal Office of Public Health, Switzerland 2021. The reference is also marked in the marked-up version of the manuscript. 

5.) We corrected the spelling of “desirability”. Thank you very much for this comment. 

6.) The uploaded version of the figures should have high resolution. This was probably not visible in the printout of the reviewers. Thanks for letting us know and we will keep an eye on it. 

7.) Thank you for this comment. We harmonized the description of the sample size, which is 510. 

Response to reviewer 2: 

Thank you very much for your nice comments about our paper. We appreciate that you liked the manuscript. Of course, we agree with the reviewer’s comment that our study analyzes only cross-sectional data. This implies that we cannot exclude the possibility that the behavior of the participants influences their attitudes versus the other way round as we assume by our hypotheses. To make this transparent and more explicit, we included the following statement in the discussion part of revised manuscript: “Since we measured attitudes and self-reported behavior at the same time, we also cannot exclude the possibility that subjects adopted their attitudes to their self-perceived behavior instead of the other way round. Hence, the assumed causal attitude-behavior link should be investigated by non-intrusive behavioral measures in combination with a time lag of the observed behavior.” We hope that this takes care of the reviewer’s concern.

---

## [Editor Report · Decision Letter 1]

8 Feb 2021

Coronavirus risk perception and compliance with social distancing measures in a sample of young adults: Evidence from Switzerland

PONE-D-20-38429R1

Dear Dr. Franzen,

We’re pleased to inform you that your manuscript has been judged scientifically suitable for publication and will be formally accepted for publication once it meets all outstanding technical requirements.

Kind regards,

Valerio Capraro

Academic Editor

PLOS ONE
---

## [Editor Report · Acceptance letter]

10 Feb 2021

PONE-D-20-38429R1 

Coronavirus risk perception and compliance with social distancing measures in a sample of young adults: Evidence from Switzerland 

Dear Dr. Franzen:

I'm pleased to inform you that your manuscript has been deemed suitable for publication in PLOS ONE. Congratulations! Your manuscript is now with our production department. 

Kind regards, 

on behalf of

Dr. Valerio Capraro 

Academic Editor

PLOS ONE